# Solver-Free Decision-Focused Learning for Linear Optimization Problems

**Senne Berden, Ali İrfan Mahmutoğulları, Dimos Tsouros, Tias Guns**
Department of Computer Science, KU Leuven
{senne.berden,irfan.mahmutogullari,dimos.tsouros,tias.guns}@kuleuven.be

## Abstract

Mathematical optimization is a fundamental tool for decision-making in a wide range of applications. However, in many real-world scenarios, the parameters of the optimization problem are not known a priori and must be predicted from contextual features. This gives rise to *predict-then-optimize* problems, where a machine learning model predicts problem parameters that are then used to make decisions via optimization. A growing body of work on *decision-focused learning* (DFL) addresses this setting by training models specifically to produce predictions that maximize downstream decision quality, rather than accuracy. While effective, DFL is computationally expensive, because it requires solving the optimization problem with the predicted parameters at each loss evaluation. In this work, we address this computational bottleneck for linear optimization problems, a common class of problems in both DFL literature and real-world applications. We propose a solver-free training method that exploits the geometric structure of linear optimization to enable efficient training with minimal degradation in solution quality. Our method is based on the insight that a solution is optimal if and only if it achieves an objective value that is at least as good as that of its adjacent vertices on the feasible polytope. Building on this, our method compares the estimated quality of the ground-truth optimal solution with that of its precomputed adjacent vertices, and uses this as loss function. Experiments demonstrate that our method significantly reduces computational cost while maintaining high decision quality.

## 1 Introduction

Linear optimization is widely used to model decision-making problems across many domains [5, 18, 20, 41]. Given a linear objective function and a set of linear constraints, an optimization solver can compute the optimal decisions. However, in many real-world scenarios, the cost coefficients that define the objective function are not known at solving time. For example, a delivery company may need to plan delivery routes without knowing the future traffic conditions. In such cases, the problem parameters must first be predicted from available contextual information (e.g., the weather conditions or temporal features). This gives rise to *predict-then-optimize* problems [26], where the predictions of a machine learning model serve as inputs to an optimization problem. The quality of the produced solutions thus hinges on the parameters predicted by the machine learning model, making the loss function used to train this model highly important.

Traditionally, the machine learning model is trained to maximize accuracy (e.g., by minimizing the mean squared error over all its predictions). However, this may lead to suboptimal decisions. This is because, while perfect predictions lead to perfect decisions, different kinds of imperfect predictions affect downstream decision-making in different ways. *Decision-focused learning* (DFL) addresses this by training models specifically to make predictions that lead to good decisions. As numerous works have shown, this generally improves decision quality [2, 9, 10, 24, 28, 33, 40].

39th Conference on Neural Information Processing Systems (NeurIPS 2025).

However, a large limitation of DFL is its high computational cost and limited scalability. This stems from the fact that gradient-based DFL involves solving the optimization problem with the predicted parameters during each loss evaluation, to assess the impact of the predictions on decision quality.

In this paper, we address this issue and greatly improve the efficiency of DFL for linear optimization problems, which have received much attention in DFL literature [2, 10, 24, 28, 26, 40], and are a mainstay in mathematical optimization. Our solver-free approach significantly reduces training time by eliminating the need for solve calls during training altogether, and hence bypassing the most computationally expensive part of the training process. It is based on the insight that a solution is optimal if and only if it achieves an objective value that is at least as good as that of its adjacent vertices on the feasible polytope. Building on this, we propose a new loss function named LAVA, that compares the quality of the ground-truth optimal solution with that of its adjacent vertices, evaluated with respect to the predicted cost vector. These adjacent vertices can be precomputed efficiently prior to training, by performing simplex pivots. Our experiments show that our solver-free LAVA loss enables efficient training with minimal degradation in solution quality, particularly on problems that are not highly degenerate.

## 2 Background

In this section, we formalize the predict-then-optimize problem setting, and give the necessary background from linear programming, which we will utilize in our methodology.

### 2.1 Predict-then-optimize

We assume a parametric linear program (LP) of the following form.

$$\min \quad c^\top z \tag{1a}$$
$$\text{s.t.} \quad Az = b \tag{1b}$$
$$z \geq 0 \tag{1c}$$

with $c \in \mathbb{R}^n$, $A \in \mathbb{R}^{m \times n}$, $b \in \mathbb{R}^m$, and $m \leq n$. We denote the set of optimal solutions as $Z^\star(c) \subseteq \mathbb{R}^n$, and a particular optimal solution as $z^\star(c) \in Z^\star(c)$. The parametric LP is written in standard form (i.e., with equality constraints), without loss of generality: if an LP is not initially in standard form, it can be converted by introducing slack variables. We further assume that the rows of the constraint matrix $A$ are linearly independent. If not, the problem includes redundant (implied) constraints that can be removed without affecting the feasible region. Finally, we assume that the feasible region $\{z \in \mathbb{R}^n \mid Az = b, z \geq 0\}$ is non-empty and bounded.

The cost parameters $c = [c_1 \ c_2 \ \dots \ c_n]$, where $c_i \in \mathbb{R}$, are unknown, but correlated with a known feature vector $x \in \mathbb{R}^d$ according to some distribution $P$. Though $P$ is not known, we assume access to a training set of examples $\mathcal{D}$, sampled from $P$. Two settings can be distinguished:

1. **Complete information**: each example in $\mathcal{D}$ is of the form $(x, c, z^\star(c))$, providing direct access to historical realizations of cost parameters $c$, and corresponding solutions $z^\star(c)$.

2. **Incomplete information**: each example in $\mathcal{D}$ is of the form $(x, z^\star(c))$, offering only the observed optimal solutions $z^\star(c)$ without revealing the underlying parameters $c$.

The incomplete information setting poses a notably more difficult learning problem than the complete information setting. The method we develop in this work does not assume access to the historical cost parameters $c$, and can thus be applied to both settings. In either setting, training set $D$ is used to train a model $m_\omega$ with learnable parameters $\omega$ – usually a linear regression model or a neural network – which makes predictions $\hat{c}$.

Unlike conventional regression, the objective in training is *not* to maximize the accuracy of predicted costs $\hat{c}$. Rather, the aim is to make predictions that maximize the quality of the resulting decisions. This is measured by the *regret*, which expresses the suboptimality of the made decisions $z^\star(\hat{c})$ with respect to true costs $c$ (lower is better).

$$Regret(\hat{c}, c) = c^\top z^\star(\hat{c}) - c^\top z^\star(c) \tag{2}$$

## 2.2 Linear programming preliminaries

We briefly review core concepts from linear programming that we use in our methodology. The feasible region $\{z \in \mathbb{R}^n \mid Az = b, z \geq 0\}$ is a convex polytope, assuming non-emptiness and boundedness. Its vertices correspond to *basic feasible solutions*, defined in the following.

**Definition 2.1** (Basis)**.** Given a matrix $A \in \mathbb{R}^{m \times n}$ with full row rank, a *basis* is a set of $m$ linearly independent columns of $A$. These columns form an invertible matrix $B \in \mathbb{R}^{m \times m}$. The variables corresponding to the columns in the basis are called *basic variables*; the remaining $n - m$ variables are called *non-basic variables*.

**Definition 2.2** (Basic solution)**.** Let $B$ be a basis. The corresponding *basic solution* is obtained by setting the non-basic variables to zero and solving $Bx_B = b$ for the basic variables (i.e., $x_B = B^{-1}b$).

**Definition 2.3** (Basic feasible solution (BFS))**.** A basic solution is called a *basic feasible solution* (BFS) if it satisfies $x_B \geq 0$. Every vertex of the feasible region corresponds to a BFS, and every BFS corresponds to a vertex of the feasible region. Thus, we use the terms interchangeably.

**Definition 2.4** (Degeneracy)**.** A BFS $z$ is said to be *degenerate* if one or more of its basic variables is zero. In such cases, multiple bases lead to the same BFS. We refer to the number of zero-valued basic variables as the *degree* of degeneracy.

For any cost vector $c$, there exists an optimal solution $z^\star(c)$ that is a BFS, and therefore, lies at a vertex of the feasible polytope. The methodology we propose in this paper will make use of the vertices *adjacent* to this optimal solution. This notion of adjacency is defined as follows.

**Definition 2.5** (Adjacent bases)**.** Two bases $B_1$ and $B_2$ are said to be *adjacent* if they differ by exactly one column; that is, the set of basic variables for $B_2$ can be obtained from that of $B_1$ by replacing a single basic variable with a non-basic one (an operation referred to as a *pivot*).

**Definition 2.6** (Adjacent vertices)**.** Two BFSs (i.e., vertices of the feasible region) are said to be *adjacent* if there exists a pair of adjacent bases such that each basis in the pair corresponds to one of the two BFS.

## 3 Related work

A lot of work on DFL has focused specifically on LPs and combinatorial problems, as handling these problems is particularly challenging. This is because they make the gradient of any loss that depends on $z^\star(\hat{c})$ (like the regret) zero almost everywhere. This can be seen when applying the chain rule:

$$\frac{\partial \mathcal{L}(z^\star(\hat{c}), c)}{\partial \omega} = \frac{\partial \mathcal{L}(z^\star(\hat{c}), c)}{\partial z^\star(\hat{c})} \frac{\partial z^\star(\hat{c})}{\partial \hat{c}} \frac{\partial \hat{c}}{\partial \omega} \tag{3}$$

The second factor $\frac{\partial z^\star(\hat{c})}{\partial \hat{c}}$ expresses the change in $z^\star(\hat{c})$ when $\hat{c}$ changes infinitesimally. However, for LPs and combinatorial problems, this factor is zero almost everywhere, and nonexistent otherwise. In other words, a small change in the problem's parameters $\hat{c}$ either does not change its solution $z^\star(\hat{c})$, or changes it discontinuously (leading to nonexistent gradients). Most work on DFL has focused on circumventing this obstacle. Three general types of approaches can be distinguished. We briefly discuss them here, but refer the reader to [26] for a comprehensive overview of the field.

The first type of approach is based on analytical smoothing of the optimization problem, in which the problem's formulation is altered such that the optimal solution changes smoothly in function of the parameters. This creates a new problem that, while only an approximation of the original, leads to non-zero gradients of the regret [25, 40]. To compute these gradients, differentiable optimization is used, which offers a way of differentiating through continuous convex optimization problems [1, 2].

The second type instead computes weight updates for the predictive model by perturbing the objective vector. Here, the difference between the solution for the predicted vector and the solution for a perturbed vector is used to compute a meaningful gradient [4, 29, 33].

The third type uses surrogate loss functions that are smooth but still reflect decision quality. The loss we propose fits this category. Other examples are the seminal SPO+ loss, as well as losses based on noise-contrastive estimation [28] and learning to rank [24]. Also [35] and [36] – which first *learn* surrogate losses, to then use them to train a predictive model – are of this type.

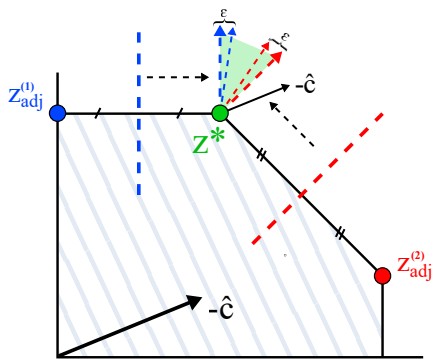

Figure 1: Optimal solution $z^\star$ has two adjacent vertices, $z_{adj}^{(1)}$ and $z_{adj}^{(2)}$. Geometrically, these vertices define the facets of the optimality cone of $z^\star$. All cost vectors $c$ for which $-c$ lies within this cone lead to $z^\star$. Cost vector $\hat{c}$ is predicted. $\hat{c}$ correctly prefers $z^\star$ to $z_{adj}^{(1)}$ (i.e., $\hat{c}^\top z^\star < \hat{c}^\top z_{adj}^{(1)}$), and thus lies on the correct side of the corresponding facet. This adds no penalty to $\mathcal{L}_{\text{AVA}}$. However, $\hat{c}$ wrongly prefers $z_{adj}^{(2)}$ to $z^\star$, and thus lies on the wrong side of the corresponding facet. This adds $\hat{c}^\top z^\star - \hat{c}^\top z_{adj}^{(2)}$ to $\mathcal{L}_{\text{AVA}}$.

Some work within DFL has instead focused on improving efficiency and scalability. Because DFL involves repeatedly solving optimization problems, training can be computationally expensive. To improve efficiency, [27] has investigated the use of problem relaxations and warm-starting techniques. Additionally, [28] and [24] have introduced losses that use a solution pool that is incrementally grown. Finally, and most directly related to this paper, [39] proposed a surrogate loss that measures the distance between the predicted cost vector and the optimality cone of the true optimal solution, which can be computed by solving a non-negative least squares problem, which is a quadratic program (QP).

In summary, most existing methods repeatedly solve the (smoothed) LP or a QP corresponding to a non-negative least squares problem at training time. Alternatively, they solve the LP to construct a surrogate loss or solution pool for training. Our approach, however, is completely solver-free, making it significantly faster than existing methods, without compromising on decision quality.

## 4 Solver-free training by contrasting adjacent vertices

We now present our approach to enable solver-free DFL for linear optimization problems. We first present a novel loss function that uses only the adjacent vertices of the optimal solutions. We then describe how these adjacent vertices can be efficiently precomputed.

### 4.1 Loss based on adjacent vertices

We construct a loss function based on the following proposition, which follows from the convexity of the feasible polytope (as proven in Appendix A).

**Proposition 4.1.** Let $z$ be a vertex of the feasible polytope, and let $Z_{adj}(z)$ be the set of vertices adjacent to $z$. Vertex $z$ is an optimal solution for cost vector $c$ if and only if $c^\top z \leq c^\top z_{adj}$ for all $z_{adj} \in Z_{adj}(z)$.

The proposition expresses that a solution is optimal if and only if none of its adjacent vertices have a better objective value. Recall that in predict-then-optimize problems, a predicted cost vector $\hat{c}$ leads to zero regret if it makes the known solution $z^\star(c)$ optimal. So, for any suboptimal cost vector $\hat{c}$ (not leading to $z^\star(c)$), one or more of the adjacent vertices will score better than $z^\star(c)$. We construct a loss – named *Loss via Adjacent Vertex Alignment* (LAVA) – that penalizes $\hat{c}$ for each such adjacent vertex.

$$\mathcal{L}_{\text{AVA}}(\hat{c}, z^\star(c)) = \sum_{z_{adj} \in Z_{adj}(z^\star(c))} \max(\hat{c}^\top z^\star(c) - \hat{c}^\top z_{adj}, -\epsilon) \quad \text{where } \epsilon \geq 0 \qquad (4)$$

The geometric interpretation of the LAVA loss is shown in Figure 1. The loss evaluates how well the predicted cost vector $\hat{c}$ distinguishes a true optimal solution $z^\star(c)$ from its adjacent vertices. Specifically, it penalizes cases where an adjacent solution $z_{adj} \in Z_{adj}$ appears more favorable than the optimal one under $\hat{c}$. Geometrically, this occurs when the predicted cost vector $\hat{c}$ falls outside the optimality cone of $z^\star(c)$, defined as the set of cost vectors for which $z^\star(c)$ remains optimal. For each adjacent vertex, the loss increases proportionally to the margin by which $c^\top z^\star(c)$ is worse than $c^\top z_{adj} - \epsilon$. Once $z^\star(c)$ is better than $z_{adj}$ by at least $\epsilon$, no further gain is enforced, reflecting the fact that, in Proposition 4.1, the precise margin by which $c^\top z \leq c^\top z_{adj}$ is irrelevant. In case there are multiple optimal solutions for ground-truth $c$ (i.e., $|Z^\star(c)| > 1$), the loss is more conservative, and is minimized only for a proper subset of cost vectors that leads to $z^\star(c)$ as optimal solution.

The value of $\epsilon$ defines a margin for how much better the optimal solution must be before further improvements are not rewarded. In principle, $\epsilon = 0$ is most in line with the optimality condition expressed in Proposition 4.1. However, after meeting the optimality condition, subsequent updates during training (e.g., from other instances) may shift the predicted cost vector slightly such that an adjacent vertex becomes preferred again. To prevent this, using a small positive value (e.g., $\epsilon = 0.1$, which we use in our experiments) introduces a buffer zone that tends to lead to smoother and more stable learning, likely by reducing sensitivity to minor fluctuations near the decision boundary. In Appendix B, we provide a visualization of how this margin parameter affects the learning curve.

The LAVA loss function has the following desirable properties:

1. It is differentiable with respect to $\hat{c}$ almost everywhere and offers non-zero gradients whenever $\hat{c}$ does not yet lead to the ground-truth optimal solution $z^\star(c)$.

2. It only requires access to ground-truth solution $z^\star(c)$, and not to the underlying true cost vector $c$. Thus, it can be applied to both the incomplete and complete information settings.

3. It is convex, ensuring that any local minimum is also a global minimum. To see why, note that both terms in $\hat{c}^\top z^\star(c) - \hat{c}^\top z_{adj}$ are affine functions of $\hat{c}$, and thus their difference is also affine (and hence convex). Taking the maximum of this affine function and constant $-\epsilon$ leads to a hinge-like function that remains convex. And since the sum of convex functions remains convex, the overall loss remains convex when taking the sum over all $z_{adj} \in Z_{adj}(z^\star(c))$.

4. It is computationally efficient to evaluate, as it does not require computing $z^\star(\hat{c})$ through a call to a solver. Instead, it only involves dot products between $\hat{c}$ and a set of precomputed solution vectors, making it highly efficient.

The LAVA loss is somewhat related to the recently proposed CaVE [39], particularly in terms of underlying objectives. The CaVE loss is the cosine distance between $\hat{c}$ and its projection onto the optimality cone of solution $z^\star(c)$. Computing the projection involves solving a QP. Instead, LAVA penalizes the "violation" of each facet of the cone separately, as illustrated in Figure 1. This requires no solve calls. Another important difference is that CaVE represents the optimality cone through its extreme rays, which correspond to the binding constraints in $z^\star(c)$. In contrast, LAVA operates on the cone's facets, each defined by the hyperplane separating $z^\star(c)$ from an adjacent vertex $z_{adj}$.

LAVA is also related to the NCE loss from [28], but whereas NCE requires an arbitrary set of feasible solutions constructed via solve calls on predicted cost vectors during training, LAVA contrasts the optimal solution with its adjacent vertices, which can be precomputed without solving. Additionally, LAVA is a hinge-like loss thresholded at $-\epsilon$, aligning with the LP optimality condition (Proposition 4.1), whereas NCE seeks to increase the difference in objective values as much as possible. This thresholding is crucial to LAVA's performance, as demonstrated in an ablation in Appendix B.

Note that LAVA is not differentiable everywhere, because each of its $\max(\hat{c}^\top z^\star(c) - \hat{c}^\top z_{adj}, -\epsilon)$ terms has a hinge-like form, which is not smooth at the kink where $\hat{c}^\top z^\star(c) - \hat{c}^\top z_{adj} = -\epsilon$. However, these kink points constitute a measure-zero set, meaning that the probability of the predicted cost vector $\hat{c}$ ending up exactly at a kink point is zero. Consequently, LAVA can be optimized reliably using standard gradient-based methods.

Also note that, although the zero vector technically lies within every optimality cone and could in principle minimize LAVA when the margin $\epsilon$ is zero, this does not occur in practice. Standard random initialization of the predictive model ensures that the predicted cost vector starts away from zero, and gradient updates induced by LAVA push predictions toward differences between optimal and adjacent solutions, rather than toward the origin. While pathological scenarios could be constructed in which

all gradients align exactly toward zero, such cases are extremely unlikely and have not been observed in practice.

We presented LAVA in the context of LPs. However, it also extends to integer linear programs (ILPs) and mixed-integer linear programs (MILPs) by operating on their LP relaxations. The effect of doing so depends on the properties of the original problem. For binary ILPs, each optimality cone in the LP relaxation is contained within the corresponding cone of the binary ILP [39]. This only makes LAVA somewhat conservative: it will always push predicted cost vectors toward the *correct* optimality cone, but it may push them further into the cone than strictly needed. For non-binary ILPs and MILPs, this containment does not necessarily hold, and the performance of using LAVA largely depends on the integrality gap of the relaxation.

## 4.2 Precomputing the adjacent vertices

---

**Algorithm 1** Find adjacent vertices of a basic feasible solution

---

1: **Input:** Basic feasible solution $z^\star$, basic indices $B(1), \ldots, B(m)$, non-basic indices
   $N(1), \ldots, N(n-m)$, constraint matrix $A$
2: **Output:** Set $Z_{adj}$ of vertices adjacent to $z^\star$
3: $B \leftarrow [A_{*B(1)} \ldots A_{*B(m)}]$
4: $N \leftarrow [A_{*N(1)} \ldots A_{*N(m)}]$
5: $Z_{adj} \leftarrow \emptyset$
6: $Queue \leftarrow [(B, N)]$
7: $Visited \leftarrow \{(B, N)\}$
8: **while** $Queue$ is not empty **do**
9:    Dequeue a basis $B_{curr}$ and corresponding $N_{curr}$ from $Queue$
10:    Compute directions of movement $D = -B_{curr}^{-1}N_{curr}$
11:    **for** each non-basic variable $z_j^\star$ as entering variable **do**
12:       $d' = D_{*j}$ is the direction of movement of the basic variables when increasing $z_j^\star$ by 1
13:       Perform minimum ratio test $\theta^\star = \min\limits_{\{i \mid d_i < 0\}} \left( -\frac{z_{B_{curr}(i)}^\star}{d_i} \right)$
14:       **if** $\theta^\star > 0$ **then**                    ▷ *Nondegenerate pivot*
15:          Let $d$ be the unit vector $e_j \in \mathbb{R}^n$
16:          **for** $k = 1 \ldots m$ **do**
17:             Set direction $d_{B_{curr}(k)} = d'_k$ for basic variable $B_{curr}(k)$
18:          Add new adjacent vertex $Z_{adj} = Z_{adj} \cup \{z^\star + \theta^\star d\}$
19:       **else**                                ▷ *Degenerate pivot*
20:          Identify $B_{curr}(i)$ as the leaving basic variable according to the TNP rule
21:          Construct new basis $B_{new}$ and corresponding $N_{new}$ by replacing $B(i)$ with $N(j)$
22:          **if** $(B_{new}, N_{new}) \notin Visited$ **then**
23:             Add $(B_{new}, N_{new})$ to $Queue$ and to $Visited$
24: **return** $Z_{adj}$

---

We outline the procedure that precomputes the adjacent vertices $Z_{adj}$ of a vertex $z^\star$ in Algorithm 1. The algorithm takes as input a basic feasible solution $z^\star$, the constraint matrix $A$, and a basis of $z^\star$ (expressed through the indices of the basic and non-basic variables). It outputs the set $Z_{adj}$ of all adjacent vertices. The algorithm starts by constructing the initial basis matrix $B$ (and the associated matrix of non-basic columns $N$), and uses these to perform pivots, as performed in the (revised) simplex method [5, 7]. Here, a pivot refers to the swapping of a basic variable and a non-basic variable. The progression through the algorithm differs depending on whether $z^\star$ is a nondegenerate or degenerate solution. We now discuss each case separately.

### 4.2.1 Base case: nondegenerate vertices

When $z^\star$ is nondegenerate, it has a unique associated basis, and every adjacent vertex can be obtained through a single pivot operation (i.e., by replacing one basic variable with a non-basic one). To do so, $D = -B^{-1}N$ is computed at line 10 of Algorithm 1. Each column $D_{*j}$ expresses how the basic variables must change to maintain $Az = b$ when non-basic variable $z_j$ is increased by 1. The algorithm then loops through the non-basic variables, and considers each $z_j^\star$ in turn as the variable

that enters the basis (line 11). This differs from the simplex method, which selects a single entering variable heuristically.

For each entering variable, the algorithm must determine the corresponding variable that should leave the basis. To do so, it computes the maximum step size $\theta^\star$ that can be taken in direction $d' = D_{*j}$ without violating the nonnegativity constraints $z \geq 0$. This is computed using the standard minimum ratio test $\theta^\star = \min_{i:d'_i < 0} \left( -z^\star_i / d'_i \right)$ [5, 7]. Since $z^\star$ is nondegenerate, $\theta^\star > 0$ will hold, and the if-block at line 14 will be entered. The corresponding adjacent vertex is now given by $z^\star + \theta^\star d$, where $d \in \mathbb{R}^n$ is obtained by setting the entry corresponding to the entering variable $j$ to 1, and setting the value of each basic variable index $B(i)$ to $d'_i$, with all other entries remaining zero (lines 15-18).

Because of nondegeneracy, the else-block at line 19 is never entered, and all adjacent vertices are found after a single iteration of the while-loop.

### 4.2.2 Edge case: degenerate vertices

In the degenerate case, multiple bases correspond to $z^\star$. This makes efficiently computing all adjacent vertices much more challenging, as they cannot all be obtained by performing pivots on any single basis $B$ of $z^\star$. In this case, some of the basis pivots performed in Algorithm 1 have a step size of $\theta^\star = 0$, and result in the same vertex $z^\star$. These pivots thus produce other bases associated with $z^\star$, which subsequently have to be explored. Thus, to ensure that all adjacent vertices of $z^\star$ are identified, one in principle needs to consider all pivot operations across all bases associated with $z^\star$. However, as shown in [12] and [23], this quickly becomes intractable.

Some work has aimed to address this, primarily between the late 1970s and early 1990s, but has since remained relatively obscure, receiving limited attention in recent research. For an overview, we refer the reader to [13, 14, 16]. Particularly relevant to this paper is the work by Geue [17] and Kruse [23], which showed that to compute all adjacent vertices of a degenerate solution, not all bases of that solution need to be explored. It suffices to only consider those bases that can be reached through lexicographic pivoting [8]. To further reduce the required number of pivots, Geue later introduced a dedicated *Transition Node Pivoting* (TNP) rule [15], which we employ in Algorithm 1. More information on the TNP rule can be found in Appendix C.

To systematically explore bases using the TNP rule, we use the *Queue* and *Visited* data structures initialized in lines 6 and 7, respectively. When an entering variable results in a step size $\theta^\star = 0$ in line 13, the algorithm applies the TNP rule to select a leaving variable (line 20). If the resulting basis is not in *Visited* yet, it is added to the *Queue* for later exploration. The *Visited* set ensures that each basis is processed only once, preventing redundant pivot operations.

### 4.3 Putting it together

Given a dataset $\mathcal{D} = \{(x, c, z^\star(c))\}$ or $\mathcal{D} = \{(x, z^\star(c))\}$, we use Algorithm 1 to find all adjacent vertices $Z_{adj}(z^\star(c))$. For each $z^\star(c)$ this will be an $k \times n$ matrix with $k$ the number of adjacent vertices. Using these matrices, we can train a predictive model $m_\omega$ over dataset $\mathcal{D}$ with $\mathcal{L}_{\text{AVA}}(m_\omega(x), z^\star(c))$.

## 5 Experimental evaluation

We evaluate our method against state-of-the-art approaches, focusing on the trade-off between training time and decision quality, and how performance scales with the number of variables and constraints.

### 5.1 Experimental setup

Here we give an overview of our experimental setup; more details can be found in Appendix D. The models are linear regressors, as is common in existing literature [10, 24, 26, 28, 34]. We train these models using the Adam optimizer [22]. All hyperparameters are tuned on independent validation sets prior to training. We train the models until their performance on the validation set has not improved by at least $1\%$ for three checks in a row, or until training time has surpassed $600$ seconds (not including evaluation on the validation set), whichever comes first. For LAVA, this training time includes both adjacent vertex precomputation and actual training. We report the test set performance of the model state that led to the best validation set performance during training. When reporting

training times, we report the time it took to reach this state. Reported results are the average taken over 5 independent runs with varying train-validation-test splits. We report the *normalized* regret:

$$Normalized\ Regret = \frac{\sum_{i=1}^{n_{test}} c^\top z^\star(\hat{c}) - c^\top z^\star(c)}{\sum_{i=1}^{n_{test}} c^\top z^\star(\hat{c})} \tag{5}$$

All experiments were conducted on a machine equipped with an Intel(R) Core(TM) i7-1165G7 processor and 16 GB of RAM. All code is implemented in Python, and builds on the PyEPO library [38]. Our code and data are available at https://github.com/ML-KULeuven/Solver-Free-DFL/.

## 5.2 Baselines

We compare with the same set of methods as used in the evaluation of [39]. These methods are:

1. **Mean Squared Error (MSE):** This method minimizes the mean squared error (MSE) between the predicted cost coefficients $\hat{c}$ and the ground-truth cost coefficients $c$. This is not a decision-focused approach, but is typically included in DFL evaluations to demonstrate the decision quality obtained when predictive accuracy is maximized during training.

2. **Smart Predict-Then-Optimize (SPO+):** This is the smart predict-then-optimize approach from [10], which minimizes the surrogate SPO+ loss, a convex upper bound of the regret that uses the true cost vector $c$. This method generally achieves great decision quality in existing comparative analyses [26, 38].

3. **Perturbed Fenchel-Young Loss (PFYL):** This method, proposed in [4], uses the Fenchel-Young losses from [6] in combination with an approach that obtains informative gradients through perturbation of the cost vectors $\hat{c}$. The size of the perturbations is controlled by hyperparameter $\sigma$.

4. **Noise-Contrastive Estimation (NCE):** This method compares the quality of the optimal solution with a set of suboptimal solutions that is gradually grown throughout training [28]. This method was introduced with the intention of improving the efficiency of DFL. For only 5% of the instances in each epoch, a solver is used to compute a solution and add it to the pool of feasible solutions; the other 95% use the pool as is.

5. **Cone-Aligned Vector Estimation (CaVE):** This method projects the predicted cost vector onto the optimality cone of the optimal solution. Doing so involves solving a non-negative least squares problem, which is a QP. The intention is that solving this QP is computationally cheaper than solving the original LP.

It is worth noting that MSE and SPO+ are given access to the true cost vectors $c$, as they are only applicable in the complete information setting. PFYL, NCE, CaVE, as well as our proposed LAVA also work in the incomplete information setting, and do not require access to $c$.

## 5.3 Benchmarks

We compare these methods on the following predict-then-optimize benchmarks, listed in increasing order of degeneracy. We refer to Appendix D for further details.

1. **Random LPs:** We construct LPs by sampling the entries of $A$ and $b$ uniformly at random, which ensures that the feasible space contains only nondegenerate vertices [19]. Each LP contains 150 variables and 50 constraints. We make sure that all generated constraints are nonredundant. As true mapping between features and cost values, we use a degree 8 polynomial that is approximated using linear regression to simulate model misspecification. This is common practice in DFL evaluations [10, 24, 34, 38, 39], and is especially relevant when training interpretable models [11, 21, 37].

2. **Multi-dimensional knapsack:** The optimization task is a multi-dimensional knapsack problem, as also considered in [25, 28, 38]. Because this problem has integer variables, we compute the adjacent vertices for LAVA on its LP relaxation, but evaluate the test set regret on the original integer problem. In the first experiment, the problem is three-dimensional and considers 300 items. In the second, we vary the problem size. Most vertices are nondegenerate, some are slightly degenerate. We use real-world data, taken from a common

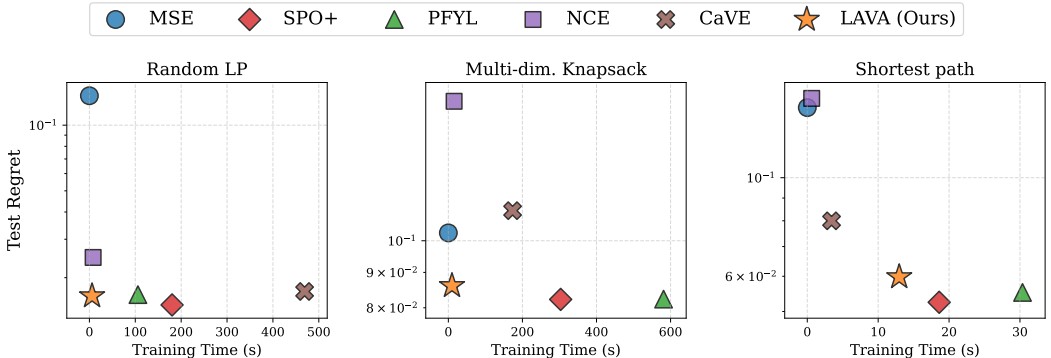

Figure 2: Comparison of the efficiency of different methods

machine learning benchmark in which the median house prices of districts in California must be predicted from 8 correlated features, including spatial features, features about the districts' populations and other aggregate housing statistics [31].

3. **Shortest path:** We include this common benchmark from the DFL literature [10, 24, 26, 28]. The optimization problem involves computing the shortest path from the bottom-left node to the top-right node of a $5 \times 5$ grid, giving rise to $40$ variables and $25$ constraints. The predicted values serve as edge costs in the grid. The true predictive mapping is the same as used in the first benchmark. All vertices are highly degenerate, making this a particularly challenging benchmark for our method.

## 5.4 Results and discussion

Figure 2 shows the training time and test set regret (log scale) of the various methods (full results with standard errors are given in tabular format in Appendix E). LAVA is Pareto-efficient on all benchmarks, meaning it is never bested in both training time and test regret by another method. On the random LPs and multi-dimensional knapsack problems, it offers a highly desirable trade-off: training time is reduced significantly with minimal degradation in solution quality compared to the best-performing methods (SPO+ and PFYL). On the highly degenerate shortest path problem, the relative test regret of LAVA is slightly worse, and the improvement in training time is less significant. Despite the shortest path being the smallest problem considered, it is the benchmark on which LAVA takes the longest.

Figure 3 shows why: the majority of LAVA's computation time is spent precomputing the adjacent vertices, which is computationally expensive due to the shortest path's high degree of degeneracy (see Section 4.2.2). Because of this, LAVA may not be the most appropriate method to address network flow problems, which tend to be highly degenerate [5]. Fortunately, these problems do not suffer much from efficiency issues in the first place, as specialized solving algorithms are available [3, 30]. Figure 3 also highlights another advantage of LAVA. In hyperparameter tuning, the adjacent vertices can be precomputed once and subsequently used in each configuration explored. Since the majority of LAVA's computation time is spent on this precomputation, the tuning process becomes particularly efficient.

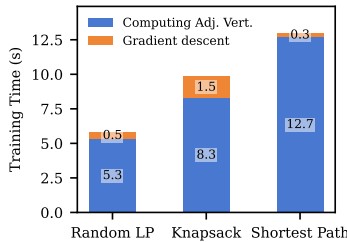

Figure 3: Breakdown of LAVA's training time

Figure 4 shows how the training time and normalized test regret scale with increasing problem size. This is shown for the multi-dimensional knapsack problem, where we vary the number of items (i.e., variables) and dimensions (i.e., constraints). The left column shows that LAVA's test regret remains on par with SPO+ and PFYL for increasing problem size. The right column shows that LAVA remains highly efficient when the problem size increases, whereas SPO+ and PFYL scale significantly worse. This is especially true for an increasing number of constraints, by which LAVA's total computation time is largely unaffected.

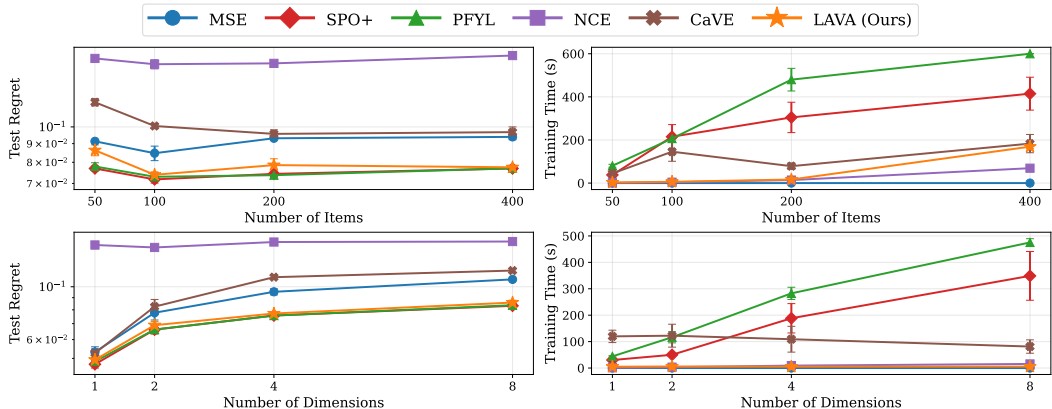

Figure 4: Comparison of performance in function of the problem size of multi-dimensional knapsack

## 6 Conclusions and future work

In this work, we significantly improve the efficiency of DFL for linear predict-then-optimize problems, by proposing a solver-free method that exploits the geometric structure of these problems. Our method, named LAVA, compares the estimated quality of the known optimal solution with that of its adjacent vertices on the feasible polytope, based on the insight that a solution is optimal if and only if it scores at least as good as its adjacent vertices. Experiments demonstrate that LAVA significantly reduces computational cost while maintaining high decision quality, and that it scales well with problem size.

Directions for future work include improving the performance of LAVA on highly degenerate problems, possibly through problem-specific adjacent vertex computation methods or heuristics. Another avenue is to investigate ways for LAVA to utilize the ground-truth cost coefficients when available (i.e., in the complete information setting). Finally, an extension of the approach to other problem classes, such as mixed-integer linear programs, through more principled methods than linear programming relaxation, is another worthwhile direction.

## 7 Acknowledgements

This research received funding from the European Research Council (ERC) under the European Union's Horizon 2020 research and innovation program (Grant No. 101002802, CHAT-Opt). Senne Berden is a fellow of the Research Foundation-Flanders (FWO-Vlaanderen, 11PQ024N).

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

# A Proof of Proposition 4.1

We start by repeating Proposition 4.1 here:

**Proposition A.1.** Let $z$ be a vertex of the feasible polytope, and let $Z_{adj}(z)$ be the set of vertices adjacent to $z$. Vertex $z$ is an optimal solution for cost vector $c$ if and only if $c^\top z \leq c^\top z_{adj}$ for all $z_{adj} \in Z_{adj}(z)$.

This is a well-known property of linear programming, and is the basis of the simplex method [5, 7]. For instance, it is stated in standard textbook [20] as 'Property 3', though without formal proof. For the sake of completeness, and to provide some intuition for why it holds, we formally prove the proposition here.

*Proof.* The forward implication holds trivially: if $z$ is an optimal solution for cost vector $c$, then $\forall z' \in \mathcal{F} : c^\top z \leq c^\top z'$, where $\mathcal{F}$ denotes the feasible region ($\mathcal{F} = \{z \in \mathbb{R}^n \mid Az = b, \ z \geq 0\}$). Since $Z_{adj}(z) \subseteq \mathcal{F}$, it holds that $\forall z_{adj} \in Z_{adj}(z) : c^\top z \leq c^\top z_{adj}$.

We must now prove the backward implication. That is, we must prove that if $\forall z_{adj} \in Z_{adj}(z) : c^\top z \leq c^\top z_{adj}$ holds, then $z$ is optimal for cost vector $c$.

We prove this by contradiction. Assume $\forall z_{adj} \in Z_{adj}(z) : c^\top z \leq c^\top z_{adj}$ holds, but that there exists a $z' \in \mathcal{F}$ for which $c^\top z' < c^\top z$.

We know that the feasible region $\mathcal{F}$ is a convex polytope. Thus, the vector $z' - z$ can be written as a conic combination of the differences $\{z_{adj} - z \mid z_{adj} \in Z_{adj}\}$:

$$\exists \lambda \geq 0 : z' - z = \sum_i^{|Z_{adj}|} \lambda_i (z_{adj} - z) \tag{6}$$

Now, since $c^\top z' < c^\top z$, it must hold that

$$c^\top (z' - z) < 0 \tag{7}$$

$$\Leftrightarrow c^\top \sum_i^{|Z_{adj}|} \lambda_i (z_{adj} - z) < 0 \tag{8}$$

$$\Leftrightarrow \sum_i^{|Z_{adj}|} \lambda_i c^\top (z_{adj} - z) < 0 \tag{9}$$

However, because of the assumption that $\forall z_{adj} \in Z_{adj}(z) : c^\top z \leq c^\top z_{adj}$, and because each $\lambda_i > 0$, each summand $\lambda_i c^\top (z_{adj} - z)$ must be nonnegative. This is a contradiction. Thus, $z$ is optimal. $\square$

# B Ablation of thresholding in LAVA

In Table 1, we show the importance of thresholding the LAVA loss at zero (i.e., $\epsilon = 0$ or using a small constant (e.g., $\epsilon = 0.1$), in comparison with removing the threshold (i.e., $\epsilon = \infty$). In Figure 5, we show the effect of using a small non-zero value for $\epsilon$ on the learning curves, using the random LP benchmark.

Table 1: Test regrets of LAVA for various thresholds

| Method | Random LP | Multi-dim. Knapsack | Shortest Path |
|---|---|---|---|
| LAVA ($\epsilon = 0$) | $0.023 \pm 0.001$ | $0.127 \pm 0.007$ | $0.061 \pm 0.008$ |
| LAVA ($\epsilon = 0.1$) | $0.016 \pm 0.001$ | $0.086 \pm 0.002$ | $0.060 \pm 0.012$ |
| LAVA ($\epsilon = \infty$) | $0.166 \pm 0.009$ | $0.156 \pm 0.003$ | $0.243 \pm 0.022$ |

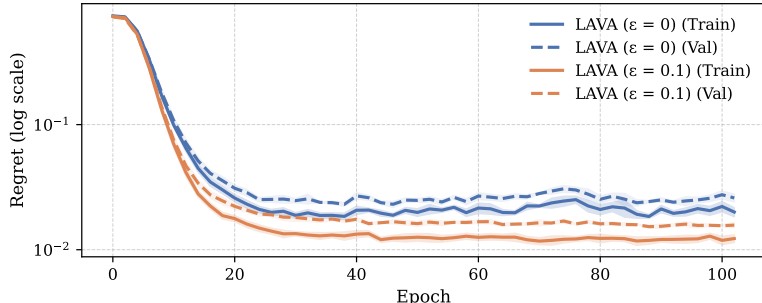

Figure 5: Learning curves for LAVA on the random LP benchmark, comparing margin parameter values $\epsilon = 0$ and $\epsilon = 0.1$.

## C   More information on the TNP rule

When a vertex $z^\star$ is degenerate, it has multiple corresponding bases. This makes efficiently computing all adjacent vertices of $z^\star$ much more challenging, as they cannot all be obtained by performing pivots on any single basis $B$ of $z^\star$. To ensure that all adjacent vertices of $z^\star$ are identified, one in principle needs to consider all pivot operations across all bases associated with $z$. However, for a $\sigma$-degenerate $n$-dimensional vertex $z^\star$ – where $\sigma$ refers to the number of zero-valued basic variables – this number of bases is lower-bounded by $U_{min} = 2^{\sigma-1}(n - \sigma + 2)$ [23], and upper-bounded by $U_{max} = \binom{n+\sigma}{\sigma}$ [12]. For instance, for $n = 100$ and $\sigma = 30$, $U_{min} = 3.865 \cdot 10^{10}$ and $U_{max} = 2.6 \cdot 10^{39}$. This makes the exhaustive exploration of all bases of $z^\star$ intractable.

This has inspired an investigation into more efficient ways of generating all adjacent vertices of a degenerate vertex – a problem which sometimes gets referred to as the *neighborhood problem* [23]. Some work has aimed to address this, primarily between the late 1970s and early 1990s, but has since remained relatively obscure, receiving limited attention in recent research. This line of work takes a graph-theoretic approach to the neighborhood problem. It associates with degenerate vertex $z^\star$ a *degeneracy graph* $G(z^\star) = (V, E)$ where $V = \{B | B \text{ is a feasible basis of } z^\star\}$ and $E = \{\{B_u, B_v\} \subseteq V \mid B_u \longleftrightarrow B_v\}$. In words, the nodes of the graph are the bases associated with $z^\star$, and an edge connects two bases if one can be obtained from the other through a single pivot operation. The nodes can be further separated into two types: internal nodes and transition nodes. *Internal nodes* correspond to bases from which all possible pivots lead to other bases associated with the same vertex $z^\star$. *Transition nodes* correspond to bases for which at least one basis of an adjacent vertex $z_{adj}$ can be reached through a pivot. This framework of degeneracy graphs has led to numerous valuable insights into various aspects of linear programming, including cycling in the simplex method, the neighborhood problem, the construction of degenerate solutions, and sensitivity analysis. For an overview, we refer the reader to [13, 14, 16].

Particularly relevant to this paper is the work by Geue [17] and Kruse [23], which showed that to compute all adjacent vertices of a degenerate solution, not all bases of that solution need to be explored. It suffices to only consider those bases that can be reached through lexicographic pivoting [8]. To further reduce the required number of pivots, Geue later introduced a dedicated *Transition Node Pivoting* (TNP) rule [15], which we employ in Algorithm 1. The TNP rule is a pivoting rule that only pivots from transition nodes to other transition nodes, and that ensures that with each pivot, at least one new basis of an adjacent vertex is discovered. This is a desirable property, as internal nodes do not contribute directly to the solution of the neighborhood problem.

To apply TNP pivoting, we must start from a transition node associated with $z^\star$. If our initial basis is not a transition node, we can obtain one by applying random pivots (with respect to an anti-cycling rule [8]). Since the starting node is now a transition node, there will be a column $d$ of $D = -B^{-1}N$ such that $\min\limits_{\{i|d_i<0\}} \left( -\frac{z^\star_{B_{curr}(i)}}{d_i} \right) > 0$, i.e., $\theta^\star > 0$ in the minimum ratio test (line 13 in Algorithm 1). This column is referred to as the *transition column*, and is denoted by $t$. This column will remain a transition column throughout all the subsequent TNP pivots. The TNP rule comes into effect when, for an entering variable with index $j$, $\theta^\star = 0$ (line 13 of Algorithm 1) and there are multiple

minimizers, i.e., $\left| I_{min}^{(j)} \right| > 1$, where $I_{min}^{(j)} = \{i \mid -\frac{z_{B_{curr}(i)}^\star}{d_i} = 0\}$. The TNP rule is used to decide which $i \in I_{min}$ to select as leaving variable. This $i$ is selected as $\max_k \{\frac{D_{kt}}{D_{kj}} | k \in I_{min}^{(j)}\}$, which ensures that $t$ remains a transition column. For a proof of this statement, as well as further details on the TNP rule, we refer the reader to [15] and [17].

## D   Further details of experimental setup

### D.1   Hyperparameters

All methods were tuned on a validation set per benchmark. The hyperparameter configuration that led to the best validation set regret was selected. Considered values for the learning rate were $0.001, 0.01, 0.1$ and $1$. For all methods, $0.01$ performed best. For PFYL, considered values for $\sigma$ were $0.01, 0.1, 1$ and $10$. For the random LPs, $0.1$ performed best. For the multi-dimensional knapsack problem and the shortest path problem, $1$ performed best. A single perturbed cost vector was sampled per backward pass. For NCE, the solve ratio was set to $5\%$. For CaVE, both the CaVE-E and CaVE+ variants were considered. On random LPs and the shortest path problem, CaVE-E performed best. On the multi-dimensional knapsack problem, CaVE+ performed best. For LAVA, $\epsilon = 0.1$ was chosen for all benchmarks.

### D.2   Optimization problems

**Random LPs:**   Random linear programs were generated in the form $\{z \in \mathbb{R}^n \mid Az \leq b, z \geq 0\}$, which were subsequently converted to standard form. The entries of $A$ were sampled uniformly from the interval $[0, 1]$. The right-hand side vector $b$ was determined in two stages. First, a solution vector $z^\star$ was sampled uniformly from $[0, 1]$. The initial entries of $b$ were then set as $Az^\star$. Subsequently, 50 randomly selected entries of $b$ were increased by a random value sampled uniformly from $[0, 0.2]$. A check was performed to ensure that none of the generated constraints were redundant (i.e., implied by other constraints). No further modifications were necessary to achieve nonredundancy. The process described above, applied to the specified problem dimensions (100 variables and 50 constraints), did not produce any redundant constraints.

**Multi-dimensional knapsack:**   Given a set of items $\mathcal{I} = \{1, \ldots, n\}$, where each item $i \in \mathcal{I}$ has a value $c_i$ and a weight $w_{ij}$ for each of $m$ knapsack constraints, the multi-dimensional knapsack problem seeks to find the subset of items with maximum total value without exceeding the capacity $W_j$ for each constraint $j \in \{1, \ldots, m\}$.

Let

$$z_i = \begin{cases} 1 & \text{if item } i \text{ is selected} \\ 0 & \text{otherwise} \end{cases}$$

for all $i \in \mathcal{I}$. The problem can then be formulated as follows:

$$\max \quad \sum_{i \in \mathcal{I}} c_i z_i \tag{10}$$

$$\text{s.t.} \quad \sum_{i \in \mathcal{I}} w_{ij} z_i \leq W_j, \quad \forall j \in \{1, \ldots, m\} \tag{11}$$

$$z_i \in \{0, 1\} \quad \forall i \in \mathcal{I}. \tag{12}$$

The objective function (10) maximizes the total value of the selected items. Constraints (11) ensure that the weight of the selected items does not exceed the capacity for each dimension $j$. The domain constraint (12) specifies that each item can either be selected in its entirety, or not at all. In generating the multi-dimensional knapsack problems, the item weights were integer values sampled uniformly at random from $[1, 10]$, and the capacity in each constraint was set to $10\%$ of the sum of the weights in that constraint.

**Shortest path problem:** This problem is taken from [10], and was also used in [24, 26, 28]. Consider a directed 5x5 grid graph $G = (V, E)$ where each node $(i, j) \in V$ is connected to its right and upward neighbors, forming a directed acyclic graph. Each edge $(i, j) \in E$ has an associated cost $c_{ij}$. The objective is to find the shortest path from the bottom-left node $(1, 1)$ to the top-right node $(5, 5)$.

Let

$$z_{ij} = \begin{cases} 1 & \text{if edge } (i, j) \text{ is included in the path} \\ 0 & \text{otherwise} \end{cases}$$

for all $(i, j) \in E$. The shortest path problem can then be formulated as the following linear program:

$$\min \quad \sum_{(i,j) \in E} c_{ij} z_{ij} \tag{13}$$

$$\text{s.t.} \quad \sum_{j: \, (1,j) \in E} z_{1j} = 1 \tag{14}$$

$$\sum_{i: \, (i,5) \in E} z_{i5} = 1 \tag{15}$$

$$\sum_{j: \, (k,j) \in E} z_{kj} - \sum_{i: \, (i,k) \in E} z_{ik} = 0, \quad \forall k \in V \setminus \{(1,1), (5,5)\} \tag{16}$$

$$\tag{17}$$

The objective function (13) minimizes the total cost of the path. Constraint (14) ensures that one unit of flow is sent out from the bottom-left node $(1, 1)$. Constraint (15) ensures that one unit of flow is received at the top-right node $(5, 5)$. The flow conservation constraints (16) maintains flow balance at each intermediate node.

Since edges only exist to the right or upwards, the set of edges $E$ can be explicitly defined as:

$$E = \{((i, j), (i + 1, j)) \mid i < 5\} \cup \{((i, j), (i, j + 1)) \mid j < 5\}$$

### D.3 True predictive mappings

**Polynomial mapping:** This mapping is commonly used in experimental evaluations of DFL methods. It was introduced in [10], and was later considered in [24, 26, 34, 38, 39].

The data generating process is the following. First, we generate the parameters of the true model as a $40 \times 5$ matrix $B$, wherein each entry is sampled from a Bernoulli distribution with probability $0.5$. We then generate features-target pairs as follows:

1. First, each element of feature vector $x \in \mathbb{R}^5$ is sampled from the standard normal distribution $\mathcal{N}(0, 1)$.

2. Then, the corresponding target value $c_j \in \mathbb{R}$ is generated as follows:

$$c_j = \frac{1}{3.5^{deg}} \left( 1 + (\frac{(B^\top x)_j}{5} + 3)^{deg} \right) \cdot \epsilon_j$$

where $\epsilon_j$ is sampled uniformly from $[1 - \bar{\epsilon}, 1 + \bar{\epsilon}]$.

In our experiments, we use $\deg = 8$ and $\bar{\epsilon} = 0$.

**California house prices:** This prediction task is taken from a common machine learning benchmark [31], which is offered as a benchmark in the scikit-learn Python package [32]. In this benchmark, the median house prices of districts in California must be predicted from 8 correlated local features, including spatial features, features about the districts' populations and other aggregate housing statistics.

Table 2: Test regrets with standard errors for different methods

| Method | Random LP | Multi-dim. Knapsack | Shortest Path |
|---|---|---|---|
| MSE | $0.136 \pm 0.003$ | $0.103 \pm 0.002$ | $0.144 \pm 0.023$ |
| SPO+ | $0.015 \pm 0.001$ | $0.082 \pm 0.001$ | $0.052 \pm 0.007$ |
| PFYL | $0.017 \pm 0.001$ | $0.082 \pm 0.001$ | $0.055 \pm 0.010$ |
| NCE | $0.025 \pm 0.001$ | $0.159 \pm 0.003$ | $0.151 \pm 0.016$ |
| CaVE | $0.017 \pm 0.001$ | $0.111 \pm 0.002$ | $0.080 \pm 0.014$ |
| LAVA (Ours) | $0.016 \pm 0.001$ | $0.086 \pm 0.002$ | $0.060 \pm 0.012$ |

Table 3: Training times with standard errors for different methods

| Method | Random LP | Multi-dim. Knapsack | Shortest Path |
|---|---|---|---|
| MSE | $0.0 \pm 0.0$ | $0.0 \pm 0.0$ | $0.0 \pm 0.0$ |
| SPO+ | $180.2 \pm 21.8$ | $303.3 \pm 74.6$ | $18.6 \pm 1.3$ |
| PFYL | $105.7 \pm 11.8$ | $581.1 \pm 8.4$ | $30.4 \pm 4.6$ |
| NCE | $8.1 \pm 0.4$ | $15.6 \pm 5.3$ | $0.6 \pm 0.3$ |
| CaVE | $469.1 \pm 37.1$ | $173.5 \pm 37.5$ | $3.4 \pm 1.1$ |
| LAVA (Ours) | $5.8 \pm 0.1$ | $9.8 \pm 1.4$ | $13.0 \pm 1.0$ |

# E    Results in tabular form

Tables 2 and 3 show the results of Figure 2 in tabular form.

