# OpenReview forum: "Solver-Free Decision-Focused Learning for Linear Optimization Problems"
_NeurIPS.cc/2025/Conference — NeurIPS 2025 poster_

### Official Review · Reviewer_qHeU · 2025-06-24

**Clarity:** 4
**Significance:** 2
**Originality:** 3
**Rating:** 5
**Confidence:** 5

**Summary:**

The method tackles improved training times for so-called predict-then-optimise problem, i.e. linear optimisation problems (LP) where the cost-function is unknown but rather needs to be predicted (an which usually require multiple LPs to be solved during training). The key technical contribution lies in a novel loss, called LAVA (loss via adjacent vertex alignment), which essentially favours clearer decision making w.r.t. the predicted cost. To this end, an optimal solution (vector), and its adjacent solution vectors are taken and the cost vector is pushed in such a direction (through back-propagation) that the decision towards the optimal solution vector is as clear as possible (i.e. the cost vector should not "lean" towards one of the adjacent solution vectors). The performance of the the proposed loss is evaluated on multiple instances of three classes of LPs.

**Questions:**

- l. 35: would be great to give examples on how suboptimal predictions can affect decision making
- l. 58 - 59 repeat of "set of optimal solution"
- l. 146-147: would be great to add that the fact that z is optimal if it has lower objective value than all its neighbours arises from the fact that the feasible region according to Ax = b is a >> convex << region
- l.203: explicitly mention that that you refer to line 10 in algorithm 1
- what is the computational complexity of algorithm 1?
- why is normalised regret used as the metric to evaluate the performance?
- who would the performance be on constrained shortest path problems (i.e. shortest path problems with additional constraints stating that "whenever this edge is taken, the other edge also has to be taken")?

**Ethical Concerns:**

["NO or VERY MINOR ethics concerns only"]

**Final Justification:**

The paper is technically solid and my main concern (i.e. solution not availble) seems to be out of scope of the problem setting. Therefore, I raised my score to "accept".

**Limitations:**

- generally, an "optimal" solution z^star might not be available, i.e. for a given dataset it might be unclear what z^star is (since the cost vector c first has to be predicted) and with that restricts the LAVA loss to the special case where this is clear which in turn poses limited applicability of the method
- for larger optimisation problems (millions of variables) it might be computationally intractable to find adjacent vertices

**Quality:**

3

**Strengths And Weaknesses:**

## Strengths
- the paper is very well written, the problem is well motivated and key concepts are explained well and backed up by illustrations
- compared to other approaches, the proposed method provides the best trade-off between training time and performance
- code is claimed to be released
- the main technical contribution is mathematically well-grounded

## Weaknesses
- the contribution of the paper seems limited to me. While the idea of the loss is well-grounded, its applicability to general predict-then-optimise problems seems not necessarily given as the optimal solution vector generally is not available (see also limitations).
- the evaluation of the loss is limited as only smaller toy examples are used. It would be interesting to see the performance on large scale LPs in which solving the LP during training would be intractable computationally

---

> ### Author Rebuttal · Authors · 2025-07-30
>
> Thank you for your review and for highlighting the clarity of our writing and the favorable trade-off our method achieves between training time and performance. We appreciate your overall positive assessment and recommendation for acceptance. Below, we address the questions posed in the review.
>
>
>
> **Points 1-4:** Thank you for pointing out these typos and suggesting some improvements to the text. We will incorporate these in the paper.
>
>
>
> **Optimal solution might not be available:** We find the described scenario somewhat unclear. Could you please elaborate on the setting you are envisioning. Generally, the optimal solutions are either assumed to be given or are precomputed using the ground-truth cost vectors prior to training, as in the following references: [4, 10, 24-28, 33-36, 38, 39].
>
>
>
> **Computational complexity of Algorithm 1:** In the base case of nondegenerate vertices, only one iteration through the while-loop at line 8 is performed. The most expensive step in the algorithm is the computation of the directions of movement (line 10), which involves computing $-B^{-1}N$, which is of complexity $O(n^3)$ for B an $n\times n$ matrix. The remaining steps only involve simple arithmetic of lower complexity, making the entire algorithm $O(n^3)$. The edge case of a degenerate vertex is more complicated. The answer to this question depends on the number of bases that would be visited by applying the TNP rule (line 20) [15, 17]. The TNP rule is a result coming from a series of works performed between the late 1970s and early 1990s (which we discuss in much more detail in Appendix C). Unfortunately, one of the latest works coming from this line of research reads “*note that the worst-case analysis with respect to the number of nodes of T-transition graphs is not yet completely finished*” [17]. We are not aware of any subsequent work that was able to provide such a worst-case analysis. The matter seems to be an open problem. The implication of this is that also the worst-case complexity of Algorithm 1 in the edge case of degenerate vertices is not known.
>
>
>
> **Why the normalised regret is used:** The (normalised) regret is a widely used performance metric in decision-focused learning [10, 24-28, 34, 38, 39]. It expresses the suboptimality of the decisions obtained using the predictions, compared to the actual optimal decisions (which would be obtained if the predictions were perfectly accurate). The normalized variant is often used as a more interpretable metric [10, 26, 34, 38, 39]. It expresses the regret in a relative manner. For example, a normalized regret of 0.1 expresses that the decisions obtained via the predictions achieve an objective value that is 10% worse than the actual optimal decisions.
>
>
>
> **Constrained shortest path problems:** We used the shortest path problem in the presented form because it is commonly used in this form in existing literature [10, 24, 26, 34, 38, 39]. We do not expect that adding additional side constraints would alter the conclusions of the evaluation.
>
>
>
> **Problems with millions of variables:** Problems with millions of variables are presently far out of reach for DFL, and are not in the scope of our work. For reference, existing work on DFL typically focuses on problems with several dozen up to several hundreds of variables. For example, benchmark datasets in the PyEPO library [38] – including shortest path, multidimensional knapsack, and travelling salesperson problems – contain 40, 32, and 190 decision variables, respectively. The Warcraft shortest path problem in [33] involves 144 variables, while the knapsack instance in [28] has 48 variables. Other problems, such as cubic top-k, web advertising, and portfolio optimization from [35] and [36], each involve 50 variables.

---

> > ### Comment · Reviewer_qHeU · 2025-08-04
> >
> > I appreciate the response of the authors which clarifies most of my concerns. While i am still concerned about the scalability of the approach I am positive about the paper.
> >
> > Response on *optimal solution not available*: I am considering scenarios where no ground-truth cost vector is available and thus the optimal solution vector cannot be computed in advance. Yet, I admit that such problems might be out of scope for the considered problems.
> >
> > Thank you again for providing more insights.

---

### Official Review · Reviewer_GG1N · 2025-06-30

**Clarity:** 3
**Significance:** 2
**Originality:** 3
**Rating:** 4
**Confidence:** 4

**Summary:**

This paper proposes a solver-free decision-focused learning (DFL) method for linear optimization problems. Specifically, it considers the prediction-then-optimize problem setting, where we need to train a machine learning model to predict the unknown parameters of an optimization problem, and then solve the optimization problem with predicted parameters.
DFL is typically computationally expensive as it requires solving the optimization problem with the predicted parameters at each loss evaluation. To address this challenge, this paper proposes a Loss via Adjacent Vertex Alignment (LAVA), which evaluates the predicted cost vector without explicitly solving the optimization problem. Based on the insight that a solution is optimal if and only if it achieves an objective value that is at least as good as that of its adjacent vertices on the feasible polytope, the proposed loss measures how the derived objective of the how well the predicted cost vector distinguishes a true optimal solution from its adjacent vertices.
Experiments demonstrate that the proposed method can achieve a favorable trade-off between computational efficiency and predictive performance.

**Questions:**

1.	Although the proposed method may achieve a favorable trade-off between computational efficiency and performance, is it possible to sacrifice training efficiency for better performance, or vice versa? If not, how could we conclude that the proposed method is better than others that are also on the Pareto frontier?
2.	It would be better to present some more real-world applications of the considered settings, especially the incomplete information setting.

**Ethical Concerns:**

["NO or VERY MINOR ethics concerns only"]

**Final Justification:**

Although LAVA still has minor limitations, such as its unability to control the trade-off between training time and decision quality and to handle MILP problems, I would like to retain my positive score.

**Limitations:**

This paper focuses on linear optimization problems. However, it seems nontrivial to extend the proposed method to harder problems like mixed-integer linear programs, as it leverages the special properties of linear optimization. Moreover, it focuses on linear regression as the predictor. These limitations may hinder its broad application.

**Paper Formatting Concerns:**

Typo:
Line 58-59: “is the set of optimal solutions”.

**Quality:**

3

**Strengths And Weaknesses:**

Strengths
1.	The presentation is good, including the problem formulation, the motivation statement, and the method illustration.
2.	The proposed method mitigates the need to solve the problem and is motivated by the theoretical property presented in Proposition 4.1. The idea makes sense and is easy to implement.

Weaknesses
1.	Although the loss function is motivated by Proposition 4.1, it would be better if the authors could present more theoretical analysis on the impact of the loss function impact on the final objective. For example, is it possible to present a theorem that if the loss is fewer than some threshold then the final objective would have a quality guarantee?
2.	According to the experiments on shortest path, the degeneracy may hinder the application of the proposed method. It would be better to conduct experiments on more datasets with degeneracy and to further investigate the relationship between the degree of degeneracy and the performance.
3.	In the experiments, only linear regression is considered as the predictor. I wonder whether stronger predictors would improve the prediction accuracy and thus improve the final objective performance. It would be better to investigate the influence of the prediction accuracy on the final performance.

---

> ### Author Rebuttal · Authors · 2025-07-30
>
> Thank you for your review and for acknowledging the clarity of our presentation and the soundness of our approach. We appreciate your positive evaluation and your recommendation for acceptance. Below, we address the questions posed in the review.
>
>
>
> **Trade-off training time and decision quality:** LAVA does not offer a direct control mechanism to balance the trade-off between training time and decision quality. However, looking at Figure 2, we would argue that its performance results in the most desirable trade-off between these two objectives. For instance, on the multi-dimensional knapsack problem, it produces only marginally worse decision quality than the state-of-the-art SPO+ approach, but is two orders of magnitude faster. Other reviewers seem to agree with this sentiment, stating that “*compared to other approaches, the proposed method provides the best trade-off between training time and performance*” (reviewer qHeU) and “*The LAVA loss is very fast to compute, differentiable, convex, and grounded in an understanding of polytope geometry, which makes it a very promising addition to the DFL arsenal*” (reviewer vtoW).
>
>
>
> **LAVA for (mixed-)integer linear programs:** For our discussion of this matter we refer to our response to reviewer vtoW. Given that three reviewers posed questions regarding the use of LAVA on (M)ILPs, this point warrants additional attention in the paper, and we will add this discussion to the camera-ready version on acceptance.
>
>
>
>
>
> **Linear predictive models:** We would like to clarify that the method is not limited to use with a linear predictive model. It can also be used with nonlinear models. The reason for the focus on linear models in the experiments is that this is standard practice in work on DFL for linear optimization problems, including in [10, 24-28, 34-36, 38, 39]. The reason for this, in turn, is that (in the complete information setting) DFL is most valuable under model misspecification, which may for instance occur when training interpretable models [11, 21, 38].
>
>
>
> **Varying degree of degeneracy:** We want to highlight that the three benchmarks included have varying degrees of degeneracy, and are ordered in increasing order of degeneracy (line 286): random LPs are nondegenerate, the multi-dimensional knapsack problem is somewhat degenerate, and the shortest path problem is highly degenerate. While including even more benchmarks with varying degrees of degeneracy would certainly have additional value, this was not possible to achieve within the page limit.

---

> > ### Comment · Reviewer_GG1N · 2025-08-04
> >
> > Thank the authors for the rebuttal. Although LAVA still has minor limitations, such as its unability to control the trade-off between training time and decision quality and to handle MILP problems, I would like to retain my positive score.

---

### Official Review · Reviewer_vtoW · 2025-06-30

**Clarity:** 4
**Significance:** 4
**Originality:** 4
**Rating:** 6
**Confidence:** 5

**Summary:**

This paper introduces a new loss called LAVA for decision-focused learning (DFL), which enables solver-free training. In DFL, the parameters of a statistical model are chosen to optimize performance on a downstream optimization task, typically a linear program (LP) or mixed-integer linear program (MILP) whose cost vector $c$ must be predicted. In this setting, the non-differentiability of the mapping from cost $c$ to (MI)LP solution $z^{\star}(c)$ is the main challenge limiting gradient-based training. While most existing methods require calling the optimization solver or a relaxation of it, LAVA instead relies on the geometric structure of the feasible set (a polytope) to evaluate the quality of a cost vector prediction $\hat{c}$. If the predicted $\hat{c}$ favors one of the neighbor vertices of $z^{\star}(c)$, then it is suboptimal and penalized as such. The resulting loss has nice theoretical properties (differentiability, convexity) and can be obtained very efficiently once the set of adjacent vertices for each optimal solution in the training set has been precomputed. Experiments on three benchmark problems against a variety of DFL algorithms demonstrate that LAVA reaches a new Pareto optimum between training time and solution quality.

**Questions:**

- Do you have any idea how to generalize this approach to the integer case? The CaVE paper discusses this aspect with their optimal subcone, does any of their insights generalize to your method?

**Ethical Concerns:**

["NO or VERY MINOR ethics concerns only"]

**Final Justification:**

My score was already maximal and the rebuttal comforted it.

**Limitations:**

Yes.

**Paper Formatting Concerns:**

None.

**Quality:**

4

**Strengths And Weaknesses:**

## Strengths

### Quality

- The LAVA loss is very fast to compute, differentiable, convex, and grounded in an understanding of polytope geometry, which makes it a very promising addition to the DFL arsenal.
- Initial experimental evaluation is very convincing.

### Clarity

- The exposition is remarkably clear and precise, as is the motivation.

### Significance

- This paper single-handedly changed my mind on solver-free approaches. My previous opinion was that they cannot work since they do not leverage the geometry of the problem, but LAVA proves that geometry awareness does not have to be mediated by solver calls. I strongly expect it to become a reference approach in this field.

### Originality

- The authors did genuinely engage with research on linear programming to precompute neighbors for degenerate vertices. This cross-fertilization between operations research and machine learning is more than welcome.
- The comparison with CaVE is enlightening: while both methods focus on aligning the objective vector with the normal cone of the optimal vertex, LAVA gets rid of the costly projection step at the cost of additional precomputation.

## Weaknesses

### Clarity

- Figure 4 is slightly hard to read, especially for colorblind readers like myself different linestyles would be helpful here.

### Significance

- My main worry about this method is whether it can truly scale to large instances, because all of the experiments are on very small problems. When facing larger instances, not only would degeneracy become much more of an issue: computational bottlenecks while solving the linear system $B^{-1} N$ might be encountered.

---

> ### Author Rebuttal · Authors · 2025-07-30
>
> Thank you for your constructive and positive review. It is especially gratifying to read that our work shifted your perspective on solver-free approaches. We appreciate your strong endorsement and are encouraged by your view that the method may become a reference in the field. Below, we address the questions posed in the review.
>
>
>
> **LAVA for (mixed-)integer linear programs:** When the optimization problem has integer variables, the adjacent vertices can be computed using the LP relaxation of the problem. The effect of doing so depends on the properties of the original problem.
>
>
>
> If the original problem is a binary ILP, then each optimality cone in the LP relaxation is a subset of the corresponding optimality cone in the binary ILP (Figure 2 in the CaVE paper [39] gives a nice illustration of this). This aspect is indeed discussed in [39], and has the same implications for LAVA. Like for CaVE, the consequence of this is only that the LAVA loss can be somewhat conservative on these problems: it will always try to push predicted cost vectors into the *correct* optimality cone, but it may push them further into the cone than they strictly need to be.
>
>
>
> If the original problem is a non-binary ILP or a mixed-integer program, then there is no guarantee that the cone in the LP relaxation is a subset of the original optimality cone. In this case, the performance of the strategy described above will largely depend on the integrality gap of the LP relaxation (see, for example, the related discussion on strong formulations of MILPs in Section 1.7 and the linear relaxation of MILPs in Section 2.2 of Laurence A. Wolsey - Integer Programming (1998, Wiley)). For problems with a small integrality gap, LAVA will likely work well. For problems with a large integrality gap, performance may be worse. For this reason, we think that exploring alternative adaptations of LAVA for non-binary ILPs and MILPs with large integrality gaps is a worthwhile direction for future work.
>
>
>
> Given that other reviewers also posed questions regarding the use of LAVA on (M)ILPs, this matter warrants additional attention in the paper, and we will add this discussion to the camera-ready version on acceptance.
>
>
>
> **Scaling with instance size:** Solving the linear system $-B^{-1}N$ has a complexity of at most $O(n^3)$ and can be done extremely efficiently with publicly available solvers. We expect that problems would have to be of very large sizes before the solving of this system would start to form a bottleneck. Even if this scenario were to occur, approximate solving methods can be used to quickly compute approximate solutions to the system. After all, even if the adjacent solutions computed would not be correct up to high numerical precision, they would likely still be accurate enough to produce informative gradients to train the predictive model with.
>
>
>
> We would also like to emphasize that the problems considered in our work are not small compared to those commonly used in the DFL literature. For example, benchmark datasets in the PyEPO library [38] – including shortest path, multidimensional knapsack, and travelling salesperson problems – contain 40, 32, and 190 decision variables, respectively. The Warcraft shortest path problem in [33] involves 144 variables, while the knapsack instance from [28] has 48 variables. Other problems, such as cubic top-k, web advertising, and portfolio optimization from [35] and [36] each involve 50 variables. In comparison, our problems contain 40, 150 and 300 decision variables. These are therefore comparable in size or larger than those used in related work.
>
>
>
> **Clarity of Figure 4:** Thank you for bringing this to our attention, and we apologize for the oversight. We will improve the legibility of Figure 4 and update it using different linestyles.

---

> ### Comment · Reviewer_vtoW · 2025-08-06
>
> Thank you, I am satisfied with these answers and will keep my score of 6. I'm excited to see how future research will draw inspiration from this work.

---

### Official Review · Reviewer_rtAP · 2025-07-02

**Clarity:** 3
**Significance:** 3
**Originality:** 3
**Rating:** 5
**Confidence:** 3

**Summary:**

- This paper proposes a novel, solve-free decision-focused learning method that eliminates the need to solve optimization problems during the learning process.
- Focusing on Linear Programs (LPs), the research leverages the property that if the feasible region is non-empty and bounded, at least one optimal solution exists at an extreme point of the polyhedron.
- Based on this property, the paper introduces a new loss function, LAVA (Loss based on Adjacent Vertices Approximation), which is derived from the difference in objective function values with extreme points adjacent to an observed solution.
- The proposed regret minimization approach offers the advantage of not requiring an optimization problem to be solved. Numerical experiments demonstrate that the proposed method enables fast learning, especially for non-degenerate problem instances.

**Questions:**

- As for the first point in the weakness I raised, the rationale behind the margin parameter's contribution to stabilizing the learning process isn't entirely clear. Could the authors explain in more detail how this margin helps stabilize learning?
- In Line 168, the authors state that LAVA is differentiable. Is this really true? LAVA is a point-wise maximum of linear functions for $\hat{c}$, so I understand that while its subgradient can be calculated, it is not generally differentiable. Please clarify if I've misunderstood.
- Can the proposed method be applied even when the observed solution is not an extreme point of the polyhedron? If an observed optimal solution lies on a facet and is not an extreme point, is it still possible to enumerate adjacent extreme points?
- Figure 4 shows that the training time of the proposed method increases more quickly when the number of items changes from 200 to 400. Might the computational time advantage of the proposed method disappear when dealing with more than 400 items?
- When training a model to predict the cost vector $c$ by minimizing LAVA, it appears that a model consistently predicting a zero vector for $c$ might be optimal. Such a model would render all feasible solutions optimal, which would be practically worthless in decision-focused learning. Is my understanding correct? Or does the proposed method naturally avoid such an undesirable model?

**Ethical Concerns:**

["NO or VERY MINOR ethics concerns only"]

**Final Justification:**

The authors have sincerely addressed all of my comments. While I recommend they revise the manuscript to incorporate these points, I believe the paper is suitable for publication.

**Limitations:**

yes

**Quality:**

3

**Strengths And Weaknesses:**

- Strengths
	- Decision-focused learning, which involves training prediction models by considering the performance of optimal solutions, is a major topic in machine learning. The computational time involved in decision-focused learning is a widely recognized practical challenge. This paper proposes an efficient learning method that leverages the polyhedral structure of LPs, showing a significant reduction in computation time for small-scale problems. This result appears to be a valuable contribution within the decision-focused learning framework.
	- Beyond demonstrating the effectiveness of the proposed method, the experiments carefully address and discuss the computational bottleneck of calculating adjacent extreme points when problems are degenerate. The experimental results presented in this research are well-argued and convincing.
	- The paper itself is well-written and easy to read.
- Weaknesses
	- The effect of introducing a margin in the proposed method isn't well understood. The paper states that introducing a margin stabilizes the training process, and Appendix B shows it reduces variance and improves the average behavior of regret. However, the reason why such a margin can contribute to stabilizing the learning process needs a more thorough discussion.
	- While it is inevitable, given the nature of the proposed method, it strongly relies on the properties of LPs. Consequently, its extensibility to more general Mixed-Integer Programs (MIPs) or convex optimization problems in a decision-focused context appears limited.

---

> ### Author Rebuttal · Authors · 2025-07-30
>
> Thank you for your thoughtful and constructive review. We are glad to read that you found our paper a valuable contribution to the decision-focused learning literature, and are pleased that you support acceptance of the paper. Below, we address the questions posed in the review.
>
>
>
> **LAVA for (mixed-)integer linear programs:** For our discussion of this matter we refer to our response to reviewer vtoW. Given that three reviewers posed questions regarding the use of LAVA on (M)ILPs, this point warrants additional attention in the paper, and we will add this discussion to the camera-ready version on acceptance.
>
>
>
> **Rationale behind margin parameter:** The margin $\epsilon$ is introduced to create a buffer zone: without it, the loss related to a single adjacent vertex becomes zero as soon as the optimal solution is of equal value or preferred. However, subsequent updates (e.g., from other training instances) may shift the predicted cost vector slightly such that this adjacent vertex becomes preferred again. The margin prevents this by requiring a stronger preference for the optimal solution. Empirically, this leads to smoother learning curves and improved decision quality at convergence. We will extend our discussion of this matter in the main text, and can add a figure comparing the learning curves with and without margin to appendix B.
>
>
>
> **Gradient vs subgradient:** We thank the reviewer for pointing this out. Indeed, each term of LAVA is a hinge-like $\max(\hat{c}^\top z^\star(c) - \hat{c}^\top z_{adj}, -\epsilon)$, which is not smooth at the kink where $\hat{c}^\top z^\star(c) - \hat{c}^\top z_{adj} = -\epsilon$. However, those kink points constitute a measure-zero set (i.e., the probability of ending up exactly at a kink point is zero), so LAVA is differentiable almost everywhere. In practice we optimize it with standard gradient methods without ever encountering nondifferentiable updates. We will update the discussion at line 168 in the paper to clarify this.
>
>
>
> **What if observed point not on a vertex:** If an optimal solution lies on a facet, one can always find at least one optimal vertex solution of the same objective value. In principle, one would first have to compute one of these optimal vertices before applying Algorithm 1. However, this scenario is unlikely in practice because (i) the simplex algorithm always terminates at a vertex and (ii) the set of cost vectors that have optimal solutions on facets of the feasible space has measure zero (i.e., is negligible).
>
>
>
> **Runtime increase going from 200 to 400 items:** We want to note that the training time increase going from 50 to 100 variables and from 100 to 200 variables are not collinear either. Admittedly, this is hard to notice on the figure, since the training times using LAVA are close to zero on the problems with less than 400 variables. This result is not entirely unexpected, however, as the computational complexity of Algorithm 1 (in the non-degenerate case) is $O(n^3)$ (see also our response to reviewer qHeU). We further investigated the runtime on the problem with 400 variables. A large portion of this time was spent precomputing the adjacent vertices of a handful of highly degenerate instances. We expect that on this problem, much faster performance with negligible degradation of decision quality can likely be achieved by, for example, treating Algorithm 1 as an anytime algorithm, and stopping the computation on highly degenerate vertices before completion after some timeout is reached. As discussed in the future work section, we consider the development of such strategies for highly degenerate vertices a promising direction for future work.
>
>
>
> **Predicting zero vectors:** This a good observation that warrants some additional discussion in the main text, which we will include. Each vertex of the feasible space has a convex polyhedral optimality cone of cost vectors that make this vertex optimal. The zero vector is indeed an element of each of these cones. Thus, in principle, predicting the zero vector is one way of minimizing LAVA (at least when margin $\epsilon$ is set to zero). However, as long as the predictive model’s parameters are initialized randomly (or in some other way that prevents the initial prediction of zero vectors), the use of LAVA is extremely unlikely to converge to this undesirable model. This can be seen as follows. Assume that there is an optimal solution $z^\star(c)$ in the training set, and that current prediction $\hat{c}$ prefers an adjacent solution $z_{adj}$ to $z^\star(c)$. This wrongful preference contributes $\hat{c}^\top z^\star(c) - \hat{c}^\top z_{adj}$ to the loss. The corresponding gradient is $z^\star(c) - z_{adj}$, i.e., a gradient descent step will 'push' the predicted cost vector $\hat{c}$ in the direction of the difference between the optimal solution and the adjacent solution. If one imagines this update applied to Figure 1 in the paper, one can see that this does not converge to the zero vector (unfortunately, we cannot provide a new figure illustrating this point, as this would violate NeurIPS’ rebuttal policy). That being said, it may be possible to manually construct pathological scenarios in which this gradient direction happens to align exactly with $-\hat{c}$ simultaneously for every instance in the training set, such that the model would converge to predicting zero vectors. However, this is extremely unlikely and not something we have ever encountered in practice.

---

> > ### Comment · Reviewer_rtAP · 2025-08-04
> >
> > Thank you for your detailed rebuttal. The authors' responses have successfully addressed my concerns. I found the discussion regarding the prediction of zero vectors particularly interesting. I will therefore maintain my positive evaluation.

---

### Decision · Program_Chairs · 2025-09-17

**Decision:**

Accept (poster)

**Comment:**

The paper proposes a new learning method for LPs that does not need an optimizer. This is done through a new loss function and is based on polyhedral geometric aspects.
Reviewers agree on good experiments, good presentation, the method being efficient and the fact that the method is solver-free is commended. On the other hand, restriction to small instances are noted. Following reviewer consensus, the paper merits acceptance at NeurIPS.